# BETTER STATE EXPLORATION USING ACTION SEQUENCE EQUIVALENCE

## ABSTRACT

Incorporating prior knowledge in reinforcement learning algorithms is mainly an open question. Even when insights about the environment dynamics are available, reinforcement learning is traditionally used in a *tabula rasa* setting and must explore and learn everything from scratch. In this paper, we consider the problem of exploiting priors about action sequence equivalence: that is, when different sequences of actions produce the same effect. We propose a new local exploration strategy calibrated to minimize collisions and maximize new state visitations. We show that this strategy can be computed at little cost, by solving a convex optimization problem. By replacing the usual $\epsilon$-greedy strategy in a DQN, we demonstrate its potential in several environments with various dynamic structures.

## 1 INTRODUCTION

Despite the rapidly improving performance of Reinforcement Learning (RL) agents on a variety of tasks (Mnih et al., 2015; Silver et al., 2016), they remain largely sample-inefficient learners compared to humans (Toromanoff et al., 2019). Contributing to this is the vast amount of prior knowledge humans bring to the table before their first interaction with a new task, including an understanding of physics, semantics, and affordances (Dubey et al., 2018).

The considerable quantity of data necessary to train agents is becoming more problematic as RL is applied to ever more challenging and complex tasks. Much research aims at tackling this issue, for example through transfer learning (Rusu et al., 2016), meta learning, and hierarchical learning, where agents are encouraged to use what they learn in one environment to solve a new task more quickly. Other approaches attempt to use the structure of Markov Decision Processes (MDP) to accelerate learning without resorting to pretraining. Mahajan & Tulabandhula (2017) and Biza & Jr. (2019) learn simpler representations of MDPs that exhibit symmetrical structure, while van der Pol et al. (2020) show that environment invariances can be hard-coded into equivariant neural networks.

A fundamental challenge standing in the way of improved sample efficiency is exploration. We consider a situation where the exact transition function of a Markov Decision Process is unknown, but some knowledge of its local dynamics is available under the form of a prior expectation that given sequences of actions have identical results. This way of encoding prior knowledge is sufficiently flexible to describe many useful environment structures, particularly when actions correspond to agent movement. For example, in a gridworld (called RotationGrid hereafter) where the agent can move forward ($\uparrow$) and rotate 90°to the left ($\curvearrowleft$) or to the right ($\curvearrowright$), the latter two actions are the *inverse* of each other, in that performing one undoes the effect of the other. During exploration, to encourage the visitation of not yet seen states, it is natural to simply ban sequences of actions that revert to previously visited states, following the reasoning of Tabu search (Glover, 1986). We observe further that $\curvearrowright\curvearrowright$ and $\curvearrowleft\curvearrowright$ both lead to the same state (represented as state 4 in Figure 1). If actions were uniformly sampled, the chances of visiting this state would be much higher than any of the others. Based on these observations, we introduce a new method taking advantage of Equivalent Action SEquences for Exploration (EASEE), an overview of which can be found in Figure 1. EASEE looks ahead several steps and calculates action sampling probabilities to explore as uniformly as possible new states conditionally on the action sequence equivalences given to it. It constructs a partial MDP which corresponds to a local representation of the true MDP around the current state. We then formulate the problem of determining the best distribution over action sequences as a linearly constrained convex optimization problem. Solving this optimization problem is computationally

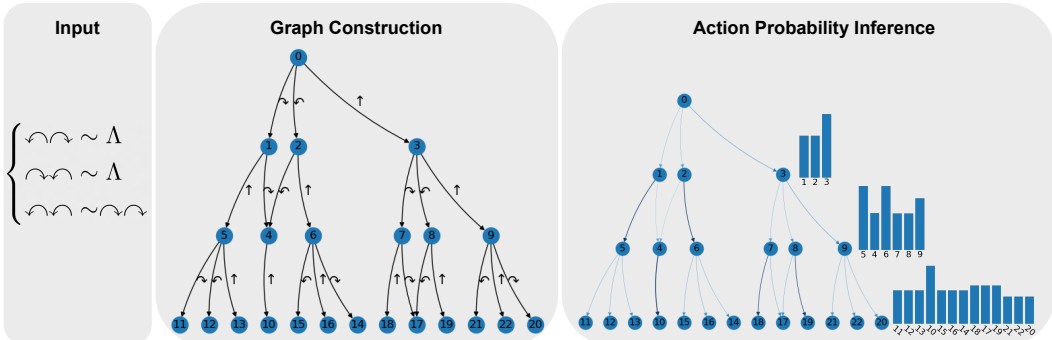

Figure 1: Illustration of EASEE on RotationGrid environment. The input is information about the dynamics of the environment known in advance under the form of action sequence equivalences ($\Lambda$ denotes the empty action sequence). This is used to construct a representation of all the unique states that can be visited in 3 steps. The probabilities of sampling each action are then determined to explore as uniformly as possible. The probabilities of visiting each unique state are displayed on the right.

inexpensive and can be done once and for all before learning begins, providing a principled and tractable exploration policy that takes into account environment structure. This policy can easily be injected into existing reinforcement learning algorithms as a substitute for $\epsilon$-greedy exploration.

Our contribution is threefold. First, we formally introduce the notion of equivalent action sequences, a novel type of structure in Markov Decision Processes. Then, we show that priors on this type of structure can easily be exploited during offline exploration by solving a convex optimization problem. Finally, we provide experimental insights and show that incorporating EASEE into a DQN (Mnih et al., 2015) improves agent performance in several environments with various structures.

**Overview** We assume that we have sets of equivalent action-sequences for the environment. Equivalent action sequences are sequences that lead to the same state. These sequences are used to build a DAG that models where the agent will end up after any sequence of actions of length $d$. Because some sequences are equivalent, several parent nodes may share a child node. A naive exploration scheme like $\epsilon$-greedy would waste resources by over exploring such child nodes. Instead, we leverage this information using the DAG constructed above; our method executes an exploratory action that maximizes the entropy of the future visited states.

## 2 RELATED WORK

**Improved Exploration** The problem of ensuring that agents see sufficiently diverse states has received a lot of attention from the RL community. Many methods rely on intrinsic rewards (Schmidhuber, 1991; Chentanez et al., 2005; Şimşek & Barto, 2006; Lopes et al., 2012; Bellemare et al., 2016; Ostrovski et al., 2017; Pathak et al., 2017) to entice agents to unseen or misunderstood areas. In the tabular setting, these take the form of count-based exploration bonuses which guide the agent toward poorly visited states (*e.g.* Strehl & Littman (2008)). Scaling this method requires the use of function approximators (Burda et al., 2019; Badia et al., 2020; Flet-Berliac et al., 2021). Unlike EASEE, these methods necessitate the computation of non-stationary and vanishing novelty estimates, which require careful tuning to balance learning stability and exploration incentives. Moreover, because these bonuses are learned, and do not allow for the use of prior structure knowledge, they constitute an orthogonal approach to ours. In Gupta et al. (2018) exploration strategies are learned from prior experience. Unlike EASEE this requires meta-training over a distribution of tasks.

**Redundancies in Trajectories** The idea that different trajectories can overlap and induce redundancies in state visitation is used in Leurent & Maillard (2020) and Czech et al. (2020) in the case of Monte-Carlo tree search. However, they require a generative model, and propose a new Bellman operator to update node values according to newly uncovered transitions rather than modifying exploration. Closer to our work, Caselles-Dupré et al. (2020) study structure in action sequences,

but restrict themselves to commutative properties. Grinsztajn et al. (2021) quantifies the probability of cycling back to a previously visited state, motivated by the analysis of reversible actions. Tabu search (Glover, 1986) is a meta-heuristic which uses knowledge of the past to escape local optima. It is popular for combinatorial optimization (Hertz & Werra, 2005). Like our approach, it relies on a local structure: actions which are known to cancel out recent moves are deemed *tabu*, and are forbidden for a short period of time. This prevents cycling around already found solutions, and thus encourages exploration. In Abramson & Wechsler (2003), tabu search is combined with reinforcement learning, using action priors. However, their method cannot make use of more complex action-sequence structure.

**Maximum State-Visitation Entropy** Our goal to explore as uniformly as possible every nearby state can be seen as a local version of the Maximum State-Visitation Entropy problem (MSVE) (de Farias & Van Roy, 2003; Hazan et al., 2019; Lee et al., 2019; Guo et al., 2021). MSVE formulates exploration as a policy optimization problem whose solution maximizes the entropy of the distribution of visited states. Although some of these works (Hazan et al., 2019; Lee et al., 2019; Guo et al., 2021) can make use of priors about state similarities, they learn a global policy and cannot exploit structure in action sequences.

**Action Space Structure** The idea of exploiting structure in action spaces is not new. Large discrete action spaces may be embedded in continuous action spaces either by leveraging prior information (Dulac-Arnold et al., 2016) or learning representations (Chandak et al., 2019). Tavakoli et al. (2018) manage high-dimensional action spaces by assuming a degree of independence between each dimension. Farquhar et al. (2020) introduce a curriculum of progressively growing action spaces to accelerate learning. These methods aim to improve the generalization of policies to unseen actions in large action spaces rather than enhancing exploration. Leveraging previous trajectories to extract prior knowledge, Tennenholtz & Mannor (2019) provide an understanding of actions through their context in demonstrations.

## 3 FORMALISM

### 3.1 EQUIVALENCE OVER ACTION SEQUENCES

We consider a *Markov Decision Process* (MDP) defined as a 5-tuple $\mathcal{M} = (\mathcal{S}, \mathcal{A}, T, R, \gamma)$, with $\mathcal{S}$ the set of states, $A$ the action set, $T$ the transition function, $R$ the reward function and the discount factor $\gamma$. The set of actions is assumed to be finite $|\mathcal{A}| < \infty$. We restrict ourselves to deterministic MDPs. A possible extension to MDPs with stochastic dynamics is discussed in Appendix A.6.

In the following, the notations are borrowed from formal language theory. Sequences of actions are analogous to strings over the set of symbols $\mathcal{A}$ (possible actions). The set of all possible sequences of actions is denoted $\mathcal{A}^\star = \bigcup_{k=0}^\infty \mathcal{A}^k$ where $\mathcal{A}^k$ is the set of all sequences of length $k$ and $\mathcal{A}^0$ contains as single element the empty sequence $\Lambda$. We use . for the concatenation operator, such that for $v_1 \in \mathcal{A}^{h_1}, v_2 \in \mathcal{A}^{h_2}, v_1.v_2 \in \mathcal{A}^{h_1+h_2}$. The transition function $T : \mathcal{S} \times \mathcal{A} \to \mathcal{S}$ gives the next state $s'$ when action $a$ is taken in state $s$: $T(s, a) = s'$. We recursively extend this operator to action sequences $T : \mathcal{S} \times \mathcal{A}^\star \to \mathcal{S}$ such that, $\forall s \in \mathcal{S}, \forall a \in \mathcal{A}, \forall w \in \mathcal{A}^\star$:

$$T(s, \Lambda) = s$$
$$T(s, w.a) = T(T(s, w), a)$$

Intuitively, this operator gives the new state of the MDP after a sequence of actions is performed from state $s$.

**Definition 1** (Equivalent sequences). *We say that two action sequences $a_1 \ldots a_n$ and $a'_1 \ldots a'_m \in \mathcal{A}^\star$ are equivalent at state $s \in \mathcal{S}$ if*

$$T(s, a_1 \ldots a_n) = T(s, a'_1 \ldots a'_m) \tag{1}$$

*Two sequences of actions are equivalent over $\mathcal{M}$ if they are equivalent at state $s$ for all $s$ in $\mathcal{S}$. This is written:*

$$a_1 \ldots a_n \sim_\mathcal{M} a'_1 \ldots a'_m \tag{2}$$

This means that we consider two sequences of actions to be equivalent when following one or the other will always lead to the same state. When the considered MDP $\mathcal{M}$ is unambiguous, we simplify the notation by writing $\sim$ instead of $\sim_{\mathcal{M}}$.

We argue that some priors about the environments can be easily encoded as a small set of action sequence equivalences. For example, we may know that going left then right is the same thing as going right then left, that rotating two times to the left is the same thing as rotating two times to the right, or that opening a door twice is the same thing as opening the door once. All these priors can be encoded as a set of equivalences:

**Definition 2** (Equivalence set). *Given a MDP $\mathcal{M}$ and several equivalent sequence pairs $v_1 \sim w_1, v_2 \sim w_2, \ldots, v_n \sim w_n$, we say that $\Omega = \{\{v_1, w_1\}, \{v_2, w_2\}, \ldots \{v_n, w_n\}\}$ is an equivalence set over $\mathcal{M}$.*

Formally, $\Omega$ is a set of pairs of elements of $\mathcal{A}^\star$, such that $\Omega \subset (\mathcal{A}^\star)^2$. By abuse of notation, we write $v \sim w \in \Omega$ if $\{v, w\} \in \Omega$.

Intuitively, it is clear that action sequence equivalences can be combined to form new, longer equivalences. For example, knowing that going left then right is the same thing as going right then left, we can deduce that going two times left then two times right is the same thing as going two times right then two times left. In the same fashion, if opening a door twice produces the same effect as opening it once, opening three times the door does the same. We formalize these notions in what follows. First, we note that equivalent sequences can be concatenated.

**Proposition 1.** *If we have two pairs of equivalent sequences over $\mathcal{M}$, i.e. $w_1$, $w_2$, $w_3$, $w_4 \in \mathcal{A}^\star$ such that*

$$w_1 \sim w_2$$

$$w_3 \sim w_4$$

*then the concatenation of the sequences are also equivalent sequences:*

$$w_1 \cdot w_3 \sim w_2 \cdot w_4$$

The proof is given in Appendix A.1. We are now going to define formally the fact that the equivalence of two sequences can be deduced from an equivalence set $\Omega$. We first consider the previous example where an action $a$ has the effect of opening a door, such that $a.a \sim a$. We can then write $a.a.a \sim (a.a).a \sim (a).a \sim a.a \sim a$ by applying two times the equivalence $a.a \sim a$ and rearranging the parentheses. More generally and intuitively, the equivalence of two action sequences $v$ and $w$ can be deduced from $\Omega$, which we denote $v \sim_{\Omega} w$, if $v$ can be changed into $w$ iteratively, chaining equivalences of $\Omega$.

More formally, we write $v \sim_{\Omega}^1 w$ if $v$ can be changed to $w$ in one steps, meaning:

$$\exists u_1, u_2, v_1, w_1 \in \mathcal{A}^\star \text{such that} \begin{cases} v = u_1.v_1.u_2 \\ w = u_1.w_1.u_2 \\ v_1 \sim w_1 \in \Omega \end{cases} \tag{3}$$

For $n \geq 2$, we say that $v$ can be changed into $w$ in $n$ steps if there is a sequence $v_1, \ldots, v_n \in \mathcal{A}^\star$ such that $v \sim_{\Omega}^1 v_1 \sim_{\Omega}^1 \cdots \sim_{\Omega}^1 v_n = w$. Finally, we say that $v \sim_{\Omega} w$ if there is $n \in \mathbb{N}$ such that $v$ can be changed into $w$ in $n$ steps. The relation $\sim_{\Omega}$ is thus a formal way of extending equivalences from a fixed equivalence set $\Omega$, and at first glance not connected with $\sim$, which deals with the equivalences of the MDP dynamics. We now show a connection between the two notions.

**Theorem 1.** *Given an equivalence set $\Omega$, $\sim_{\Omega}$ is an equivalence relationship. Furthermore, for $v, w \in \mathcal{A}^\star$, $v \sim_{\Omega} w \Rightarrow v \sim w$.*

The proof is given in Appendix A.2. Given this relation between $\sim$ and $\sim_{\Omega}$, we will simplify the notation in what follows by writing $\sim$ instead of $\sim_{\Omega}$ when the equivalence set considered is unambiguous. As $\sim_{\Omega}$ is an equivalence relationship, it provides a partition over action sequences: two action sequences in the same set lead to the same final state from any given state.

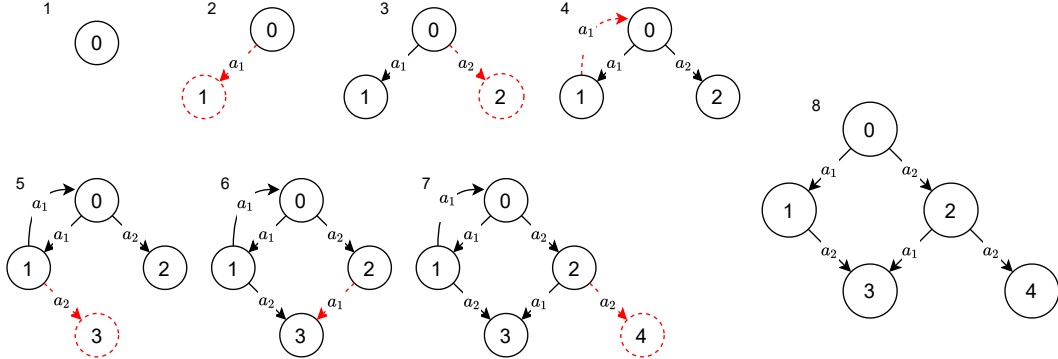

Figure 2: Example of iterative graph construction with $\Omega = \{a_1 a_1 \sim \Lambda, a_2 a_1 \sim a_1 a_2\}$ and a maximum depth of 2. The $8^{\text{th}}$ construction step corresponds to the pruning of the edge $(1, 0)$.

### 3.2 LOCAL-DYNAMICS GRAPH

We leverage the equivalences defined above to determine a model of the MDP up to a few timesteps. As traditionally done in Monte-Carlo Tree Search (Coulom, 2007), an MDP $(\mathcal{S}, \mathcal{A}, T, R, \gamma)$ with deterministic dynamics can be locally unrolled to produce a tree, where a node of depth $h$ represents a sequence of actions $v \in \mathcal{A}^h$, and the edges represent transitions between such sequences. The root of the tree corresponds to the empty action sequence $\Lambda$. Here we adopt the same formalism, except that equivalent sequences will point to the same node.

Given a tree $\mathcal{T}$ of depth $d \in \mathbb{N}$ corresponding to a partial unrolling of sequences in $\mathcal{A}^\star$, and an equivalence set $\Omega$, we call *local-dynamics graph* of depth $d$ under equivalence $\Omega$ the graph $\mathcal{G} = (V, E)$ corresponding to the tree $\mathcal{T}$ where nodes are quotiented with the equivalence relation $\sim_\Omega$. Intuitively, it means that nodes corresponding to equivalent action sequences are merged. In this case, the resulting graph is not necessarily a tree. In the following, unless the distinction is necessary, we identify action sequences with their equivalence classes.

The graph $\mathcal{G}$ gives rise to a new, smaller MDP resulting from $\mathcal{M}$: the state space $V$ is the set of action sequences smaller than $d$ quotiented by the equivalence relation $\sim_\Omega$, the action space $\mathcal{A}$ is untouched. Given a node $n$ corresponding to a sequence $w \in \mathcal{A}^\star$, and an action $a \in \mathcal{A}$, $T(n, a)$ is the node representing the sequence $w.a \in \mathcal{A}^\star$. Nodes representing sequences of length exactly $d$ are *final states*. The initial state $v_0$ is the empty sequence $\Lambda$. This MDP represents the local dynamics induced by $\sim_\Omega$ from a given root state. We detail in the next section how to construct such graphs in practice, and how to use these sub-MDPs for a better exploration.

## 4 EQUIVALENT ACTION SEQUENCES FOR EXPLORATION (EASEE)

### 4.1 FROM EQUIVALENT ACTIONS TO LOCAL-DYNAMICS GRAPH

Producing the local-dynamics graph involves considering all possible action sequences and merging those that are equivalent. Figure 2 illustrates the construction of a local-dynamics graph, given $\mathcal{A} = \{a_1, a_2\}$ and $\Omega = \{a_1 a_1 \sim \Lambda, a_2 a_1 \sim a_1 a_2\}$. Starting from the root node 0 (first step), we iteratively expand the graph by unrolling the nodes at the edges of the graph. Steps 2 and 3 create nodes 1 and 2 corresponding to action sequences $a_1$ and $a_2$ respectively. In a tree, the expansion of a node corresponding to a sequence $w \in \mathcal{A}^h$ with the action $a \in \mathcal{A}$ always leads to the creation of a new leaf that results from the sequence of actions $w.a \in \mathcal{A}^{h+1}$. However, in a local-dynamics graph the node representing $w.a$ might already be present, in which case we add an edge from $w$ without creating a new node. As a final construction step, we prune edges which go backward in the local-dynamics graph, like $(1, 0)$ in Fig. 2, such that the resulting graph is a DAG. This is motivated by the fact that we are interested in finding a good exploration policy: an action which takes us back to a previously visited state should be ignored.

From a practical point of view, the graph construction algorithm takes as input the action set $\mathcal{A}$, the sequence equivalence set $\Omega$, and the desired depth $d$, and outputs a DAG. Informally, it starts from a graph $\mathcal{G} = (V, E)$ reduced to a root state $\{0\}$ and iteratively expands $\mathcal{G}$ until a distance $d$ to the root is reached. We store in each node every action sequence which allows to reach it from any parent nodes. When expanding a node $n$ with an action $a \in \mathcal{A}$, we check every sequence $w$ stored in $n$ if $w.a$ appears in $\Omega$, and if a node corresponding to an equivalent sequence of $w.a$ is already in $V$. If it is the case, we simply add an edge from $n$ to this node, otherwise we create a new node representing $w.a$. We provide a more detailed implementation of this algorithm in Appendix A.3.

**Proposition 2.** *The complexity of this graph construction algorithm is upper bounded by* $O\left(|\mathcal{A}|^{2d}|\Omega|d\right)$.

The proof is given in Appendix A.4. It is to be noted that this upper bound is in general far larger than the actual number of operations. Indeed, it supposes that the number of nodes in the graph is $|\mathcal{A}|^d$, although it can be much smaller thanks to the redundancies induced by $\Omega$. A more precise formula is $O\left(|V||\mathcal{A}|^d|\Omega|\right)$, where $|V|$ is the number of nodes in the final graph and depends on the structure of $\Omega$. Despite this exponential theoretical complexity, the goal is to use this algorithm locally, thus for small depths. In practice we found that local-dynamics graphs could be computed within a few seconds on a standard laptop.

## 4.2 FROM LOCAL-DYNAMICS GRAPH TO LOCAL EXPLORATION POLICY

Once the local-dynamics graph $(V, E)$ has been constructed, our goal is to find a good local exploration policy in the resulting MDP as defined in Section 3.2. We recall that its set of states is $V$, and its actions dynamics are given by the edges $E$. Ideally, we would want to find a policy $\pi$ such that all nodes in the local-dynamics graph are visited equally often.

Given a policy $\pi$, a state $v \in V$ and an action $a \in \mathcal{A}$, we denote $p_{\pi,t}(v)$ and $p_{\pi,t}(v, a)$ the $t$-steps state distribution and state-action distribution respectively. Formally, $p_{\pi,t}(v) = \mathbb{P}_\pi(v_t = v)$ and $p_{\pi,t}(v, a) = \mathbb{P}_\pi(v_t = v, a_t = a)$.

Ideally we would like each $t$-step state distribution to be uniform. However, depending on the exact local-dynamics graph this may or may not be possible (see Figure 1 for an example where obtaining a uniform distribution is impossible). Instead, following the principle of maximum entropy (Jaynes, 1957), we frame the objective of balancing the state distribution at step $t$ as maximizing $H([p_{\pi,t}(v_0), p_{\pi,t}(v_1), \ldots, p_{\pi,t}(v_{|V|-1})] = H(p_{\pi,t})$, where $H$ is the Shannon entropy. For a local-dynamics graph of depth $d \in \mathbb{N}$, we define our global objective as maximizing $J(\pi) = \tilde{J}(p_{\pi,1}, \ldots, p_{\pi,d}) = \frac{1}{d} \sum_{t=1}^{d} H(p_{\pi,t})$. Other global objectives are possible, for example optimizing entropy over only the final states, or some other weighted mixture. In practice, over simple experiments, we observed that changes in the entropy mixture hardly induced any variation in the computed policies and agent behavior.

Informally, our objective can be understood as maximizing state diversity locally, for every timestep smaller than $d$. For environments where additional priors about state interests are available, one could adapt the quantity $J$ to compute the entropy on a subset of the most interesting states, therefore biasing exploration toward promising areas.

We consider $\mathcal{K}$, the set of joint distributions $(p_0, p_1, \ldots p_d)$ which verifies the following properties:

- $\forall t \leq d, p_t(v, a) \geq 0$
- $\forall v \in V, \sum_{a \in \mathcal{A}} p_0(v, a) = p_0(v) = \mathbb{1}_{v_0}(v)$
- $\forall t < d, \sum_{a \in \mathcal{A}} p_{t+1}(v, a) = \sum_{v' \in V, a \in \mathcal{A}} p_t(v', a)\mathbb{P}(v \mid v', a)$

We denote $D(\mathcal{A})$ the set of distributions over $\mathcal{A}$. From any $(p_0, p_1, \ldots, p_d) \in \mathcal{K}$, it is possible to find a time-dependent policy $\pi : V \times \{0, \ldots, d\} \to D(\mathcal{A})$ such that $p_0 = p_{\pi,0}, p_1 = p_{\pi,1}, \ldots, p_d = p_{\pi,d}$, and for any policy $\pi$ we have $(p_{\pi,0}, p_{\pi,1}, \ldots, p_{\pi,t}) \in \mathcal{K}$ (Puterman, 2014).

As the entropy $H$ is concave, the function $\tilde{J}$ is a concave function over $\mathcal{K}$. Moreover, the constraints defining $\mathcal{K}$ are linear. Therefore,

$$\max_{(p_1,\ldots,p_d)\in\mathcal{K}} \tilde{J}(p_1, \ldots, p_n) \tag{4}$$

can be solved efficiently using any convex solver. In our implementation, we use CVXPY (Diamond & Boyd, 2016; Agrawal et al., 2018). Once $(p_1^\star, \ldots, p_d^\star) = \arg\max_\mathcal{K} \tilde{J}$ is computed, we can immediately calculate a time-dependent policy $\pi^\star$ from such a distribution (Puterman, 2014) with:

$$\pi_t^\star(v, a) = \frac{p_t^\star(v, a)}{p_t^\star(v)} \qquad (5)$$

As the local-dynamics graph $(V, E)$ is a DAG, the set of nodes $V_0, V_1, \ldots, V_d$ which can be reached respectively at timesteps $t = 0, t = 1, \ldots, t = d$ are disjoint. Therefore any time-dependent policy defined on $V$ can be framed as a stationary policy. Considering for example $\pi^\star$, we can write $\pi^\star(v, \cdot) = \pi_0^\star(v, \cdot)$ if $v \in V_0$, $\pi^\star(v, \cdot) = \pi_1^\star(v, \cdot)$ if $v \in V_1, \ldots$, and $\pi^\star(v, \cdot) = \pi_d^\star(v, \cdot)$ if $v \in V_d$.

### 4.3 FROM LOCAL EXPLORATION TO GLOBAL POLICY

The optimal $\pi^\star$ determined in the previous section can then be used to guide exploration. With an $\epsilon$-greedy policy, each step has a probability $\epsilon$ of being an exploration step, where an action is sampled uniformly. Instead, we keep in memory the local-dynamics graph, and initialize the current state at $v = \Lambda$. Everytime an action $a$ is performed, $v$ is updated such that $v \leftarrow v.a$, and reinitialized to $\Lambda$ after a sequence of length $d$. At each exploration step, instead of sampling $a$ uniformly, EASEE samples $a$ according to the distribution $\pi^\star(v, \cdot)$. Pseudocode for this process can be found in Appendix A.5.

## 5 RESULTS

For every experiments, additional details about environments and hyperparameters are given in Appendix B.

### 5.1 PURE EXPLORATION

To get a better understanding of EASEE, we consider two simple gridworld environments with different structures: CardinalGrid and RotationGrid. These environments are both $100 \times 100$ gridworlds, but with different action structures. In CardinalGrid, the agent can move one square in the four cardinal directions $(\rightarrow, \leftarrow, \uparrow, \downarrow)$, whereas in RotationGrid, the agent can move either forward one square $(\uparrow)$, or rotate $90°$ on the spot to the left $(\curvearrowleft)$ or to the right $(\curvearrowright)$. The agent starts in the middle of the grid and can explore for 100 timesteps, after which the environment is reset.

In CardinalGrid, we consider the 4 equivalence sets:

- $\{\rightarrow\leftarrow\sim\leftarrow\rightarrow\}$ (" $\rightarrow$ and $\leftarrow$ commute ")
- $\{\rightarrow\leftarrow\sim\leftarrow\rightarrow, \uparrow\downarrow\sim\downarrow\uparrow\}$ ("all actions commute")
- $\{\rightarrow\leftarrow\sim\leftarrow\rightarrow, \uparrow\downarrow\sim\downarrow\uparrow, \rightarrow\leftarrow\sim \Lambda\}$ ("all actions commute and $\rightarrow\leftarrow\sim \Lambda$")
- $\{\rightarrow\leftarrow\sim\leftarrow\rightarrow, \uparrow\downarrow\sim\downarrow\uparrow, \rightarrow\leftarrow\sim \Lambda, \uparrow\downarrow\sim \Lambda\}$ (" all actions commute and $\rightarrow\leftarrow\sim\uparrow\downarrow\sim \Lambda$ "),

while in RotationGrid, we consider the three equivalence sets:

- $\{\curvearrowright\curvearrowleft\sim \Lambda\}$
- $\{\curvearrowright\curvearrowleft\sim \Lambda, \curvearrowleft\curvearrowright\sim \Lambda\}$
- $\{\curvearrowright\curvearrowleft\sim \Lambda, \curvearrowleft\curvearrowright\sim \Lambda, \curvearrowright\curvearrowright\sim\curvearrowleft\curvearrowleft\}$.

Fig. 3 shows the benefits of exploiting the structure of the action space for exploration. Figures 3a, 3b show the ratio of the number of unique states visited using EASEE over a standard uniform exploration policy. For both environments, a greater equivalence set leads to a more efficient exploration. In the environmentCardinalGrid for example, for a fixed depth of 6, adding the information that $\rightarrow$ and $\leftarrow$ commute and that every actions commute allow to reach respectively 10% and 60% more states in 100 episodes. Furthermore, extra equivalences encoding that $\rightarrow$ is the inverse of $\leftarrow$, and $\uparrow$ the inverse of $\downarrow$ increase the number of new states encountered threefold. It can also be seen that deeper graphs provide better exploration, which is expected: using deeper graphs results in exploiting equivalence priors over longer action sequences.

Figures 3c, 3d show the number of unique states visited with respect to the total number of episodes of exploration. We see that EASEE benefits exploration in all configurations considered: it allows the agent to visit more states within a single trajectory, and as well as across a thousand. It gives insight about the sample-efficiency gain which can be achieved using EASEE over a standard random policy. In the CardinalGrid setting, EASEE visits more unique states over 100 episodes than uniform exploration over 1000.

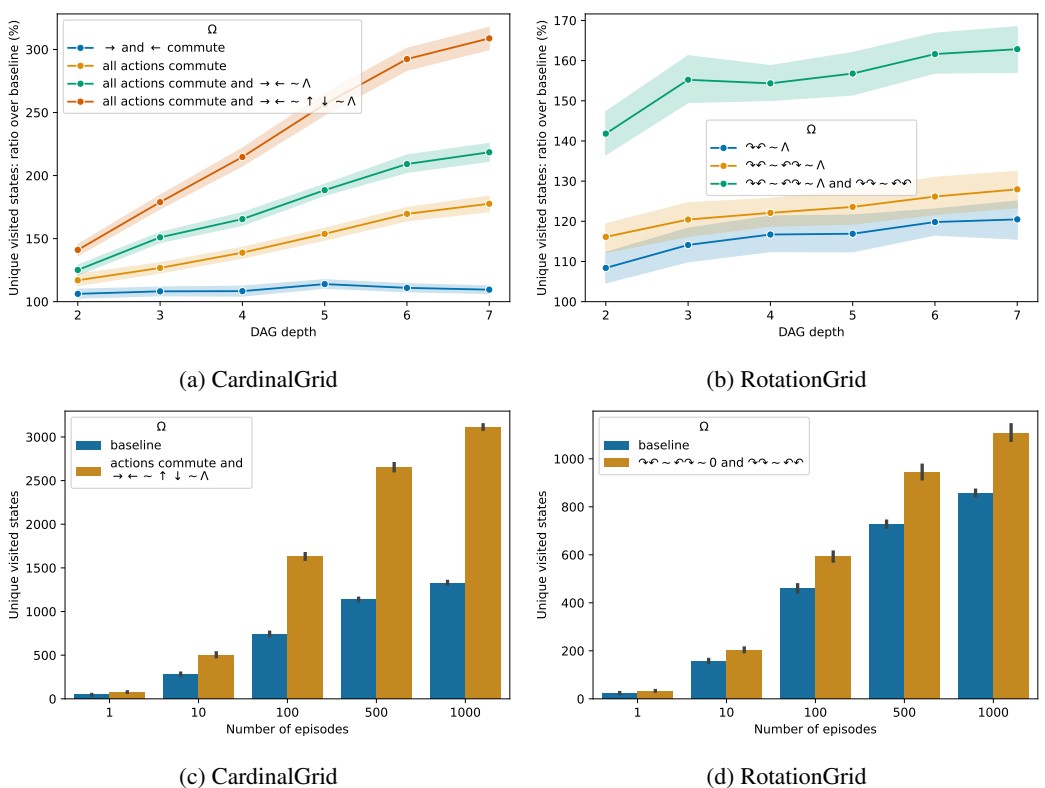

Figure 3: **(a, b):** Ratio of the number of unique visited states during 100 episodes following EASEE over standard $\epsilon$-greedy policy, for different equivalence sets and depths in the environments CardinalGrid and RotationGrid respectively. **(c, d):** Number of unique visited states according to the number of episodes for EASEE with a fixed depth of 4 compared to standard $\epsilon$-greedy policy.

## 5.2 MINIGRID

The Minimalistic Gridworld Environment (MiniGrid) is a suite of environments that test diverse capabilities in RL agents (Chevalier-Boisvert et al., 2018). We evaluated the influence of adding EASEE to Q-learning on the DoorKey task. The environment is a gridworld split into two rooms separated by a locked door. The agent must collect a key to get to the objective in the other room.The dynamics of the environment are those of RotationGrid with two extra actions: the agent may `PICKUP` the key when facing it and `OPEN` the door when carrying the key. The EASEE version of the Q-learning assumes the following action sequence equivalences:

$$\curvearrowright\curvearrowright \sim \Lambda$$
$$\curvearrowleft\curvearrowleft \sim \Lambda$$
$$\curvearrowleft\curvearrowleft \sim \curvearrowright\curvearrowright$$
$$\text{OPEN} \sim \text{OPEN} \cdot \text{OPEN}$$
$$\text{PICKUP} \sim \text{PICKUP} \cdot \text{PICKUP}$$

The reward over this training is presented in Figure 4a. Using a depth of 6, the EASEE augmented version outperforms classic Q-learning.

## 5.3 CATCHER

We test EASEE on a game of Catcher, where the agent must catch a ball falling vertically with a paddle that can move left and right. It receives a reward of $+1$ when the ball is caught and $-1$ when it is missed. The prior we incorporate into the exploration is that the actions commute *i.e.* $\leftarrow \rightarrow \sim \rightarrow \leftarrow$. For faster learning we restrict each episode to a single ball drop, with the agent starting in the middle of the environment.

We choose a depth of $30$ for EASEE. This is also the length of a single episode. The mean reward over training is plotted in Figure 4b.

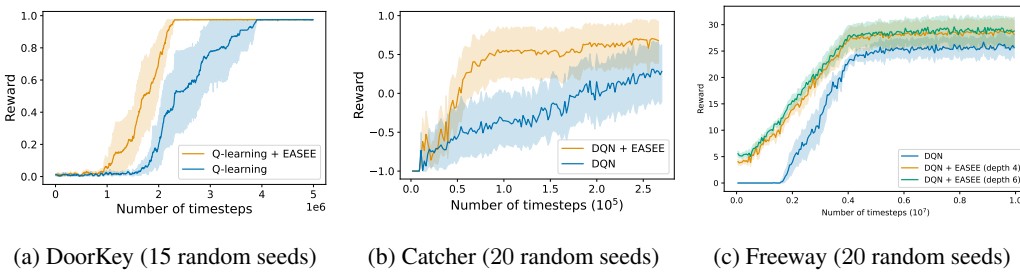

(a) DoorKey (15 random seeds)    (b) Catcher (20 random seeds)    (c) Freeway (20 random seeds)

Figure 4: Mean reward over training with $95\%$ confidence intervals.

## 5.4 FREEWAY

We test our method on the Atari 2600 game Freeway (Bellemare et al., 2013). To illustrate EASEE's performance in a non-deterministic setting, we add stochasticity to the dynamics of the game using sticky actions (Machado et al., 2018). The agent has to cross a road with multiple lanes without getting hit by the cars, and receives a reward when it reaches safety on the other side. The action space is composed of 3 actions : moving forward of 1 lane ($\uparrow$), moving backward of 1 lane ($\downarrow$), and passing ($-$). As cars arrive randomly, it is not easy to find priors on action equivalences in this environment. Since passing and moving backwards can sometimes be useful to avoid cars we cannot forbid these actions. However, we have prior knowledge that performing these two actions does not lead to visiting new lanes. We restrict the use of these actions with $\Omega = \{\downarrow \sim \downarrow \downarrow, -\downarrow \sim \downarrow -\}$, which has the effect of removing every node which is reached by chaining two $\downarrow$ actions without moving forward, and compute the exploration policy on the remaining nodes. Results can be seen in Fig. 4c.

## 6 DISCUSSION

We assume that implementers of reinforcement learning agents can provide insights about the environment, despite not knowing its precise dynamics or optimal policy.

In this work, we argue that some of these insights can be efficiently represented using the notion of action-sequence equivalence, which we formalize. We propose a method to incorporate such priors in classic Q-learning algorithms and demonstrate empirically its ability to improve sample efficiency and performance. More precisely, our approach can be divided into two steps: first, the construction of a graph representing the local dynamics, and then the resolution of a convex optimization problem aiming to balance node visitation. We show that incorporating such prior knowledge can replace standard $\epsilon$-greedy and improve at little cost RL algorithms, which traditionally start from a *tabula rasa* setting, learning everything from scratch.

We expect EASEE to be robust to slight errors in the action sequence equivalence set. It may be that the states at the end of two sequences are not exactly the same but very similar, or that an action-sequence equivalence is verified at all but a few states. In such cases, the exploration policy we determine is not optimal, but should be much closer to optimality than uniformly sampling actions.

We additionally experimented with EASEE on two other Atari games, where the improvements are less pronounced: the details are given in Appendix B.4, as well as possible explanations.

## 7 REPRODUCIBILITY STATEMENT

In an effort to help with reproducibility, we provide a light-weight implementation of the experiments discussed in sections 5.1 and 5.2. For Catcher and Atari environments, all details of the environments, as well as the preprocessing steps and the hyperparameters we chose are described in Appendix B.2 and Appendix B.3 respectively. Additionally, the pseudo-code for the proposed methods is given in Appendix A.3.

## 8 ETHICS STATEMENT

A direct application of this paper is to make reinforcement learning techniques in general, and deep Q networks in particular, more efficient at solving Markov decision processes by making use of prior knowledge about action sequence equivalences. On its own, we do not expect EASEE to have immediate societal risks. Moreover, approaches incorporating prior knowledge in reinforcement learning algorithms often increase sample efficiency at little cost, thereby leading to less expensive and more environmentally-friendly methods.

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

## A  TECHNICAL ELEMENTS AND PROOFS

### A.1  PROOF OF PROPOSITION 1

**Proposition 1.** *If we have two pairs of equivalent sequences over $\mathcal{M}$, i.e. $w_1$, $w_2$, $w_3$, $w_4 \in \mathcal{A}^\star$ such that*

$$w_1 \sim w_2$$

$$w_3 \sim w_4$$

*then the concatenation of the sequences are also equivalent sequences:*

$$w_1 \cdot w_3 \sim w_2 \cdot w_4$$

*Proof.* For any $s \in \mathcal{S}$, we have $T(s, w_1) = T(s, w_2)$ as $w_1 \sim w_2$. We apply the same property for $w_3$ and $w_4$ on the state $T(s, w_1)$:

$$T(T(s, w_1), w_3) = T(T(s, w_2), w_4)$$
$$T(s, w_1.w_3) = T(s, w_2.w_4)$$

Therefore $w_1.w_3 \sim w_2.w_4$. □

### A.2  PROOF OF THEOREM 1

**Theorem 1.** *Given an equivalence set $\Omega$, $\sim_\Omega$ is an equivalence relationship. Furthermore, for $v, w \in \mathcal{A}^\star$, $v \sim_\Omega w \Rightarrow v \sim w$.*

**$\sim_\Omega$ is an equivalence relation**   Let $u, v, w \in \mathcal{A}^\star$.

*Proof.*

- We immediately have $v \sim_\Omega^1 v$ by choosing $v_1 = \Lambda$ in equation 3, and therefore $v \sim_\Omega v$, thus $\sim_\Omega$ is reflexive.

- It is clear from its definition that $\sim_\Omega^1$ is symmetric, as $\sim$ is symmetric. Then, we suppose that $v \sim_\Omega w$. We have $n \in \mathbb{N}$ and $v_1, \dots, v_n \in \mathcal{A}^\star$ such that $v \sim_\Omega^1 v_1 \sim_\Omega^1 \cdots \sim_\Omega^1 v_n \sim_\Omega^1 w$, therefore $w \sim_\Omega^1 v_n \sim_\Omega^1 \cdots \sim_\Omega^1 v_1 \sim_\Omega^1 v$, thus $w \sim_\Omega v$. Hence $\sim_\Omega$ is symmetric.

- If $u \sim_\Omega v$ and $v \sim_\Omega w$, we have $n_1, n_2 \in \mathbb{N}$, and $u_1, \dots, u_{n_1} \in \mathcal{A}^\star$, $v_1, \dots, v_{n_2} \in \mathcal{A}^\star$, such that $u \sim_\Omega^1 u_1 \sim_\Omega^1 \cdots \sim_\Omega^1 u_{n_1} \sim_\Omega^1 v$ and $v \sim_\Omega^1 v_1 \sim_\Omega^1 \cdots \sim_\Omega^1 v_{n_2} \sim_\Omega^1 w$. It is then clear that $u \sim_\Omega^1 u_1 \sim_\Omega^1 \cdots \sim_\Omega^1 u_{n_1} \sim_\Omega^1 v \sim_\Omega^1 v_1 \sim_\Omega^1 \cdots \sim_\Omega^1 v_{n_2} \sim_\Omega^1 w$, and thus $u \sim_\Omega w$. Therefore $\sim_\Omega$ is transitive.

The relation $\sim_\Omega$ is reflexive, symmetric and transitive. Therefore it is an equivalence relation. □

**$\sim_\Omega$ implies $\sim$**

*Proof.* Let $v, w \in \mathcal{A}^\star$. From Proposition 1, we immediately get $v \sim_\Omega^1 w \Rightarrow v \sim_\mathcal{M} w$. Then we can prove by immediate induction that $\forall n \in \mathbb{N}, v_1, \dots, v_n \in \mathcal{A}^\star, v \sim_\Omega^1 v_1 \sim_\Omega^1 \cdots \sim_\Omega^1 v_n \sim_\Omega^1 w \Rightarrow v \sim w$, from which we deduce $v \sim_\Omega w$ implies $v \sim w$. □

---

**Algorithm 1:** Graph Construction

---

**Input** Action set $A$;
**Input** Equivalence set $\Omega$;
**Input** Maximum tree depth $d$;
Initialize the graph $\mathcal{G} = (V, E)$ with $V = \{0\}$ and $V = \emptyset$ ;
Initialize the set of states to expand $\mathcal{S} = \{0\}$ ;
Initialize the current tree depth $l = 0$;
Initialize a dictionary $\mathcal{E}$ which stores partial sequences of $\Omega$ for each state of $V$ ;
**while** $l < d$ **and** $\mathcal{S} \neq \emptyset$ **do**
   newStates = {} ;
   **for** *each state in $\mathcal{S}$* **do**
      **for** *each action in $A$* **do**
         `/* create a node corresponding to` $T(state, action)$ `*/`
12         newState = `expandNode`(state, action, $\Omega$, $\mathcal{E}$) ;
         **if** *newState not in $V$* **then**
            `/* Because of sequence redundancies, the state may`
               `already appear in the graph.` `*/`
            $V \leftarrow V \cup \{\text{newState}\}$ ;
            newStates $\leftarrow$ newStates $\cup \{\text{newState}\}$ ;
         **end**
         $E \leftarrow E \cup \{(\text{state, newState})\}$ ;
         `/* Update the equivalences` $\mathcal{E}$`(newState) to account for`
            `the new ways of reaching newState` `*/`
18         $\mathcal{E} \leftarrow$ `UpdateDic`(newState, $\mathcal{E}$, $\Omega$) ;
      **end**
   **end**
   $l \leftarrow l + 1$;
   $\mathcal{S} \leftarrow$ newStates ;
**end**
`/* Prune edges such that the resulting graph is a DAG.` `*/`
$T$ = `GraphToDAG`($\mathcal{G}$) ;
**Output** DAG $\mathcal{G}$;

---

## A.3 GRAPH CONSTRUCTION ALGORITHM

We present in Algorithm 1 an overview of the graph construction algorithm. It takes as input the action set $A$, the sequence equivalence set $\Omega$, and the desired depth $d$, and outputs a DAG. Informally, it starts from a graph $\mathcal{G} = (V, E)$ reduced to a root state $\{0\}$ and iteratively expands $\mathcal{G}$ until a distance $L$ to the root is reached. For a node $n \in V$ we store in $\mathcal{E}(v)$ sequences which reach $v$, and are prefixes of sequences of $\Omega$. When expanding a state $v \in V$ using an action $a \in A$ (Line. 12), we look at every partial sequence $s \in \mathcal{E}(v)$. If $s.a$ is in $\Omega$, it means that we have found a redundant sequence. If the equivalent sequence has already been computed, it means that a node $u$ representing $T(v, s.a)$ has previously been added in $\mathcal{G}$. Otherwise, we add a new node $u$. In both case, we update the equivalences $\mathcal{E}(u)$ to account for the new ways of reaching $u$ (Line. 18).

## A.4 GRAPH CONSTRUCTION COMPLEXITY

As shown in Section 3.2, constructing the graph necessitates three intricate loops: The first one goes over every internal node $n \in V$, the second one loops over the set of actions $\mathcal{A}$, and the last one loops over every partial sequence which allows to reach $v$ from a parent node. Inside these three loops, one has to compare the partial sequence with every sequence of $\Omega$. As sequence length in $\Omega$ can be bounded by $d$, the complexity cost inside the tree loops is bounded by $O(|\Omega|d)$. The total complexity is therefore lower than $O(|V||\mathcal{A}||\mathcal{A}|^{d-1}|\Omega|d) = O(|V||\mathcal{A}|^d|\Omega|d)$. As $|V| \leq |\mathcal{A}|^d$, the complexity can also be bounded by $O(|\mathcal{A}|^{2d}|\Omega|d)$.

## A.5 MODIFIED DQN

Our modified version of the DQN algorithm can be found at Algorithm 2.

---

**Algorithm 2:** Modified DQN

---

Initialize replay memory $\mathcal{D}$ and $Q$-networks $Q_\theta$ and $Q_{\theta'}$;
Determine local-dynamics graph $\mathcal{G}$ and the associated optimal exploration policy $\pi^\star$;
**for** *episode = 1 to M* **do**
    Initialize new episode;
    **for** *t = 1 to T* **do**
        $\epsilon \leftarrow$ set new $\epsilon$ value with $\epsilon$-decay ($\epsilon$ usually anneals linearly or is constant);
        Initialize at empty sequence $v \leftarrow \Lambda$;
        **if** $\mathcal{U}([0,1]) < \epsilon$ **then**
            Sample exploring action $a_t \sim \pi^\star(v, \cdot)$;
        **else**
            Select greedy action $a_t$;
        **end**
        $v \leftarrow v.a_t$ (append $a_t$ to the end of sequence $v$);
        Execute $a_t$ and observe next state $s_{t+1}$ and reward $r_t$ ;
        Store $(s_t, a_t, r_t, s_{t+1})$ in replay buffer $\mathcal{D}$ Update $\theta$ and $\theta'$ normally with minibatches
          from replay buffer $\mathcal{D}$;
        **if** *Length(v)=d* **then**
            Reset $v \leftarrow \Lambda$;
        **end**
    **end**
**end**

---

## A.6 POSSIBLE EXTENSION TO THE STOCHASTIC CASE

In this section we discuss the possibility of extending EASEE to the case of MDPs with stochastic transitions. EASEE relies on three components: the formalization of action sequence equivalences (Def. 1), the construction of a local-dynamics graph (Section 3.2), and the construction of a local exploration policy by solving a convex problem (Section 4.2). We now detail for each step the necessary changes to adapt EASEE to $\mathcal{M} = (\mathcal{S}, \mathcal{A}, T, R, \gamma)$, a MDP with stochastic dynamics.

- **Action Sequence equivalences**: the difference with the deterministic case here is that given an action $a \in \mathcal{A}$ and a state $s \in \mathcal{S}$, $T(s, a)$ is not a state but a distribution over the set of states $\mathcal{S}$. Therefore every equality considered in Section 3.1 has now to be understood not as an equality between two states but between two distributions. Other than this the formalism can be kept identical. Intuitively, two sequences of actions are equivalent if they lead to the same state distribution from any given state, *i.e.* if they produce the same effect everywhere.

- **Local-Dynamics Graph**: Here, the formalism can again be kept identical. A node in the local-dynamics graph will not represent a state anymore, but rather a distribution over $\mathcal{S}$.

- **Local Exploration Policy**: Solving directly the objective given in equation 4 would lead to maximize the diversity among state distributions encountered. As is, it would not necessarily lead to a better diversity among states, as two different distributions can have an almost similar support. Therefore, adapting EASEE to a stochastic setting would require encoding additional priors about the distributions represented by the nodes of the local-dynamics graph, which we leave for future work. If we suppose that the distributions encountered have disjoint supports, and that their entropy is the same, EASEE can be applied without modification.

## B  EXPERIMENTAL DETAILS

### B.1  GRIDWORLDS

We tested EASEE on the DoorKey task. An illustration of the initial state is given in Fig. 5. The agent is represented by the red triangle. The yellow key is necessary to open the yellow door. The two room are respectively $12 \times 17$ and $4 \times 17$ grids. The agent has $3249$ timesteps to reach the goal and receive a reward of $1$ before the environment is reset.

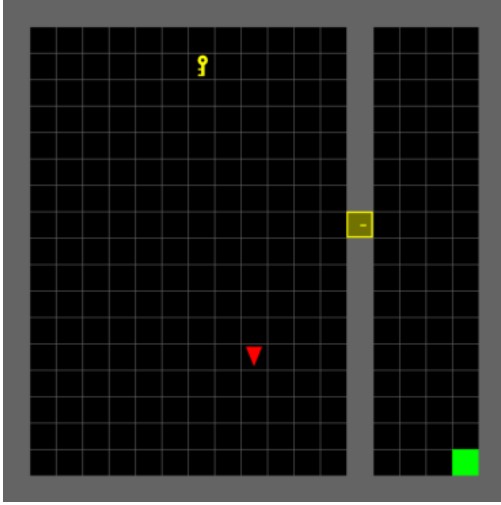

Figure 5: Example of initial state of DoorKey environment.

### B.2  CATCHER

The paddle is 1 block wide. The environment is 60 blocks wide and 30 blocks high. The ball and the paddle both move at a rate of 1 block per timestep, so each episode lasts 30 timesteps.

We use the same architecture for the DQN with and without EASEE. Each observation is a $60 \times 30$ image. The feature extractor network is a CNN composed of 3 convolution layers with kernel size 3 followed by ReLU activation. In both cases, we update the online network every 4 timesteps, and the target network every $10^3$ timesteps. We use a replay buffer of size $10^4$, and sample batches of size 32. We use the Adam optimizer with a learning rate of $10^{-4}$.

We train for $3.10^5$ timesteps. The exploration parameter $\epsilon$ is linearly annealed from 1 down to 0.05 over 20% of the training period. Other DQN hyperparameters were defaults in Raffin et al. (2019).

## B.3 FREEWAY

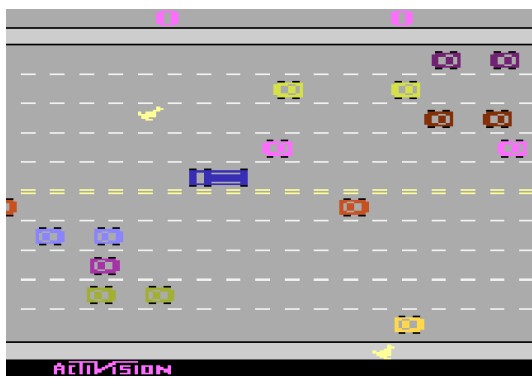

Figure 6: The Freeway environment from Atari 2600.

**Environment**    In Freeway, the agent has to cross a road with multiple lanes without getting hit by the cars. It only receives a reward when it safely reaches the other side of the road. An illustration is given in Fig. 6. The agent is represented by the yellow chicken.

To add stochasticity to the dynamic of the environment, we use sticky actions as proposed in Machado et al. (2018), with a stickiness parameter of 0.25. More precisely, the environment has 0.25 probability of executing the previous action again instead of the current desired action. The frame is recast as a $84 \times 84 \times 3$ image, and the number of frames to skip between each observation is set to 4. The reward is scaled to [-1, 1]. An observation corresponds to 4 stacked game frames.

**Architecture and hyperparameters**    We use the same architecture for the DQN with and without EASEE. Input images first go through a convolutional neural network, with the same architecture as in Mnih et al. (2015). We update the online network every 4 timesteps, and the target network every $10^3$ timesteps. We use a replay buffer of size $10^5$, and sample batches of size 32. We use the Adam optimizer with a learning rate of $10^{-4}$.

We train for $10^7$ timesteps. The exploration parameter $\epsilon$ is linearly annealed from 1 down to 0.01 over 10% of the training period, which are the default in Raffin (2018) for Atari games. Other DQN hyperparameters were defaults in Raffin et al. (2019).

## B.4 ADDITIONAL EXPERIMENTS

**Environments**    We experimented EASEE on two other Atari environments, where the action sequence structures are less straight-forward. The three environments are preprocessed as explained in AppendixB.3.

- **Boxing**: This game shows a top-down view of two boxers. The player can move in all four directions, and punch his opponent (pressing the "`FIRE`" button). The action space is composed of 18 actions : `NOOP, FIRE, UP, RIGHT, LEFT, DOWN, UPRIGHT, UPLEFT, DOWNRIGHT, DOWNLEFT, UPFIRE, RIGHTFIRE, LEFTFIRE, DOWNFIRE, UPRIGHTFIRE, UPLEFTFIRE, DOWNRIGHTFIRE, DOWNLEFTFIRE`. We incorporated priors by decomposing actions, in the form of `UPRIGHT` $\sim$ `UP.RIGHT`, `UPLEFT` $\sim$ `UP.LEFT`, `UPRIGHTFIRE` $\sim$ `UPRIGHT.FIRE`, `UPLEFTFIRE` $\sim$ `UPLEFT.FIRE`, *etc...*
- **Carnival**: The goal of the game is to shoot at targets, which include rabbits, ducks, owls, scroll across the screen in alternating directions, and sometimes come at the player. The player can only move in 1 direction, such that the action space is composed of 6 actions: [`NOOP, FIRE, RIGHT, LEFT, RIGHTFIRE, LEFTFIRE`]. As `NOOP` is not an useful action, EASEE could get an edge simply by adding the equivalence `NOOP` $\sim$ $\Lambda$. For a fair

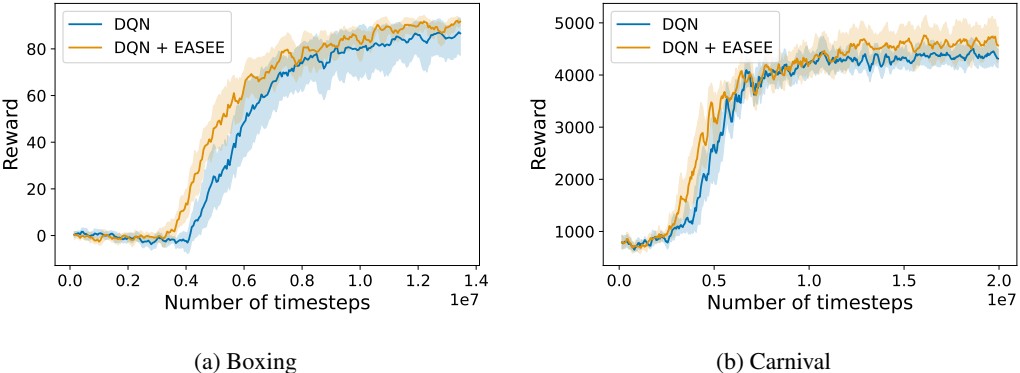

(a) Boxing

(b) Carnival

Figure 7: Performances of DQN and DQN + EASEE on the Atari 2600 games Boxing, Carnival. A 95% confidence interval over 10 random seeds is shown.

comparison, we restricted the action space to meaningful actions by removing `NOOP` for both EASEE and the baseline. We limited ourselves to the commutative property of `RIGHT` and `LEFT`: `RIGHT.LEFT` $\sim$ `LEFT.RIGHT`.

**Architecture and hyperparameters** We use the same architecture and the exact same DQN parameters as in AppendixB.3. In all three environments, EASEE is used with a depth of 4.

**Results** We can see the results on Fig.7. We can see a slight gain for Boxing, and a marginal improvement for Carnival. This can come from various factors:

- When the number of action sequence equivalences considered is small compared to the number of actions, as it is the case for Carnival, the exploration policy computed with EASEE is very much like a uniform policy. It logically makes its performances converge toward those of a standard DQN.
- The action sequence equivalences considered here are only approximately true. In boxing, it is only approximately true that `UPRIGHT` $\sim$ `UP.RIGHT` for example. In Carnival, `RIGHT` and `LEFT` commute as long as the player is not at the edges of the screen, in which case `RIGHT` or `LEFT` could have no effect. In both cases, this induces a bias that may harm performance.

