# OpenReview forum: "Better state exploration using action sequence equivalence"
_ICLR.cc/2022/Conference — ICLR 2022 Submitted_

### Official Review · Reviewer_FrAt · 2021-10-24

**Correctness:** 4
**Technical Novelty And Significance:** 4
**Empirical Novelty And Significance:** 2
**Recommendation:** 8
**Confidence:** 4

**Main Review:**

The idea proposed in the paper is quite elegant, albeit its realization is quite complex. The paper does a great job at explaining all the steps that go from equivalent sequences of actions, to an actual local policy that can be queried for an exploratory action. Overall, the paper is well-written, even if its complexity is sometimes a bit difficult for the reader.

I think that the paper would greatly benefit from a quick summary of what it does. In the introduction, the last paragraph gives a bit of guidance, but already a bit too technical in my opinion ("Then, we show that priors on this type of structure can easily be exploited during offline exploration by solving a convex optimization problem") for instance. I would put, either in the introduction or in a dedicated "Overview" section, an intuition of the steps that the algorithm takes:

- Assume that we have sets of equivalent actions sequences for the environment. Equivalent action sequences are sequences that lead to the same state (please correct me if I'm wrong)
- These sequences are used, from a current state $s_t$, to build a DAG, that sorts of models where the agent will end up after any sequence of actions of length $d$. Because some sequences are equivalent, several parent nodes may share a child node. This is the information that we want to extract.
- In a Q-Learning-like algorithm, every time-step, construct the DAG above for $s_t$, then, with probability $\varepsilon$, execute an exploratory action $a_t$ that maximizes the entropy of the future visited states.

A further intuition about the fact that the resulting exploratory policy will, basically, assign low probabilities to actions that share a same future outcome, may also help the reader understand why the proposed algorithm is beneficial to exploration.

Something else that could be made clearer is when the state-space has to be discretized. It seems that the DAG nodes don't consider states, but it is not that clear to me, and I have the feeling that explicitly mentioning what happens to the states in the Freeway example may help.

Other than that, I think that the paper is highly novel and proposes a complicated (but justified) algorithm. As such, I recommend accepting it.

**Summary Of The Paper:**

The paper proposes a method that, from a simple encoding of sequences of actions that have equivalent outcome in an MDP, allows to compute a local policy for local high-quality exploration (it replaces the random action of $\varepsilon$-greedy with an action that maximizes the entropy of future visited states). Full algorithmic details about how to do that are given in the paper, and experiments on a few environments show that the method is promising. The Freeway experiment is particularly interesting, as it shows that a (small) benefit can be obtained from the method even on an Atari game, with minimal domain knowledge (only a few equivalent action sequences are encoded), and using a slightly modified DQN algorithm.

**Summary Of The Review:**

The paper is relatively well-written, and proposes a sound algorithm that leads to good empirical results (even if the amount of empirical results is quite low). The algorithm is general enough to be applicable to an Atari game, with minimal engineering effort. The approach is quite novel, and multi-disciplinary (RL, information theory, convex optimization). As such, I think that the paper will be very interesting to people attending ICLR, and recommend accepting it.

---

> ### Author Response · Authors · 2021-11-23
> **Official comment**
>
> We thank the reviewer for the positive feedback on the proposed method. We provide additional remarks and clarifications in what follows.
>
> Q: I think that the paper would greatly benefit from a quick summary of what it does. In the introduction, the last paragraph gives a bit of guidance, but already a bit too technical in my opinion ("Then, we show that priors on this type of structure can easily be exploited during offline exploration by solving a convex optimization problem") for instance. I would put, either in the introduction or in a dedicated "Overview" section, an intuition of the steps that the algorithm takes.
>
> A: Thank you for your suggestion. We have added an overview at the end of the introduction.
>
> Q: Something else that could be made clearer is when the state-space has to be discretized. It seems that the DAG nodes don't consider states, but it is not that clear to me, and I have the feeling that explicitly mentioning what happens to the states in the Freeway example may help.
>
> A: It is perfectly right that DAG nodes are an abstract and discretized representation of the environment states. In the case of Freeway, which we modified during the rebuttal to add stochasticity in the dynamics using sticky actions [2], nodes would represent state distributions. But such (possibly complex) distributions fortunately do not need to be made explicit in order to apply EASEE: only the collisions between them are taken into account.
>
> [2]: Revisiting the Arcade Learning Environment: Evaluation Protocols and Open Problems for General Agents, Machado et al., IJCAI 2018.

---

### Official Review · Reviewer_5h9r · 2021-11-01

**Correctness:** 4
**Technical Novelty And Significance:** 2
**Empirical Novelty And Significance:** 2
**Recommendation:** 5
**Confidence:** 4

**Main Review:**

The paper is written very clearly and the contributions are easy to follow. The idea of using explicit equivalences is fairly simple and I found it surprising this has not been done before (or at least, I couldn't find such a reference). I was thinking about how this work could be meaningfully built upon to improve a general zero-knowledge RL algorithm. I see only a single extension, where the exploration strategies are learned: however this is already studied in prior work [1]. The paper therefore seems to fill a "void" in the hand-crafted case with known priors. I don't find this sufficiently significant as its own contribution. Additionally, the applicability seems to be limited to grid-like MDPs, where it is easy to hand-craft the action-equivalences. For more general environments the improvement seems marginal (Figure 7).
A suggestion how to make the work and empirical results more interesting: perhaps the problem can be cast as a Bayesian update using the hand-crafted priors with meta-learning [1].

Other notes:

- "Ideally we would like each t-step state distribution to be uniform. However, depending on the exact local-dynamics graph this may or may not be possible." Why? Can you give an example when this would not be possible? I believe it should be straightforward to use uniform strategy in the abstracted MDP and translate it back to the original problem. Since this is much simpler than solving the optimization problem, it would be preferrable, and a comparison to the max-entropy solution should be made.
- Action Space Structure: there is no explanation how the last 3 references relate to the work.


[1] Meta-Reinforcement Learning of Structured Exploration Strategies, Abhishek Gupta, Russell Mendonca, YuXuan Liu, Pieter Abbeel, Sergey Levine


**Summary Of The Paper:**

In the context of reinforcement learning, authors propose an exploration strategy based on environment-specific prior knowledge of action equivalence. An example of such equivalence is rotating 180° twice in a grid world, as the agent comes back to the same original state: the action sequence forms an identity in this case. The proposed exploration, instead of picking a random action with probability (\epsilon)/(number of actions), builds an abstracted lookahead tree of specified depth that merges equivalent action sequences and adjusts the distribution of \epsilon over them. Authors demonstrate this i) increases the number of unique state visitations in grid-world examples, and ii) DQN with their exploration strategy achieves a higher reward faster than the baseline.

**Summary Of The Review:**

I recommend 5) marginally below the acceptance threshold. There is existing literature that tackles the problem of exploration in a more general way without handcrafting prior knowledge into the problem.

---

> ### Author Response · Authors · 2021-11-23
> **Official comment**
>
> We thank the reviewer for their comments and additional references. We provide additional remarks and clarifications in what follows.
>
> Q: I see only a single extension, where the exploration strategies are learned: however this is already studied in prior work [1].
>
> A: Thank you for the additional reference that we added in the paper (Related Work: Improved Exploration). Although the objective of improving exploration is shared between [1] and our work, we feel the approaches differ significantly in spirit. Meta-training and prior knowledge assume very different access to resources and understanding regarding the environment. In many cases it may not be practical or feasible to develop the suite of similar tasks necessary for meta-learning. Moreover, [1] focuses on policy-based methods while our work is more readily used with value-based methods.
>
> Q: Additionally, the applicability seems to be limited to grid-like MDPs, where it is easy to hand-craft the action equivalences. For more general environments the improvement seems marginal (Figure 7).
>
> A: We complexified the environment Freeway by adding “sticky actions” (setting from [2]): with probability 0.25, the agent executes the previous action instead of the current desired action, which makes the dynamics stochastic. Our method worked out-of-the-box without the need of modifying any hyperparameters.
> This shows that EASEE can perform well in complex environments. Thus, the poor performances in Fig.7 are likely due to poor prior knowledge, rather than the inherent complexity of these environments.
>
> Q: "Ideally we would like each t-step state distribution to be uniform. However, depending on the exact local-dynamics graph this may or may not be possible." Why? Can you give an example of when this would not be possible?
>
> A: An example is provided in Figure 1: if the 2-step distribution is uniform there is a ⅙ chance of sampling each of the states 7, 8, and 9. Since states 7,8, and 9 are only accessible via state 3, this means that there must be a ½ chance of sampling state 3. But if the 1-step distribution is uniform then there is a ⅓ chance of sampling state 3.
>
> Q: I believe it should be straightforward to use uniform strategy in the abstracted MDP and translate it back to the original problem. Since this is much simpler than solving the optimization problem, it would be preferable, and a comparison to the max-entropy solution should be made.
>
> A: In cases where a uniform strategy exists in the abstract MDP, it could indeed be computed easily with a level-wise inductive approach, and it would lead to the same solution that the one computed with entropy-maximization. However, as shown in the previous answer, it is not necessarily the case. Thus, finding a better policy requires oversampling some actions and undersampling others in a way that is related to the local dynamics structure.
>
> Q: Action Space Structure: there is no explanation how the last 3 references relate to the work.
>
> A: These three papers make use of the structure of the action space to speed-up the learning procedure (one with curriculum, one with demonstrations, one with a self-supervised approach). We made the link with our method clearer in the paper.
>
> [2]: Revisiting the Arcade Learning Environment: Evaluation Protocols and Open Problems for General Agents, Machado et al., IJCAI 2018.

---

> > ### Comment · Reviewer_5h9r · 2021-11-26
> > **Response to comment**
> >
> > Thank you for the response and a revision of the paper. Now I understand the computation of the exploration distribution clearly.
> >
> > I still maintain that this is a nice paper that fills a gap, but I find it is not significant enough for ICLR. The number of domains where there are known equivalences seems limited and there is limited future work that could further improve on these results.

---

### Official Review · Reviewer_Ln9a · 2021-11-04

**Correctness:** 4
**Technical Novelty And Significance:** 3
**Empirical Novelty And Significance:** 2
**Recommendation:** 5
**Confidence:** 4

**Main Review:**

Strengths:

-The paper formalizes an interesting phenomenon of equivalent action sequences that exists in some MDPs.

-The method to construct a sampling policy that maximizes entropy is interesting. It uses a local dynamics graph built using action sequence equivalence knowledge  + convex optimization to  accomplish this.


Weaknesses:

-The formulation of equivalent action sequences could be simplified -- although nice, I’m not sure all of the formality is necessary here.

-The method is designed for deterministic MDPs, in which case it is believable that the transitions are known. The authors also discuss an extension to the stochastic case, where equivalences are determined based on next state distributions. This seems nearly impossible in the model-free RL setting. Even in a dynamic programming setup where the stochastic transitions are fully known, it can be difficult to verify equality of distributions. Moreover, even if it were possible to verify, I cannot think of any problem where we’d expect equality of next state distributions between action sequences.

-All experiments are performed on simple domains. It is unclear how applicable this method is in real problems.


**Summary Of The Paper:**

This paper considers a property of MDPs where multiple sequences of states end up at the same action. The main contributions of this paper are (1) a formalization of the action sequence phenomenon, (2) an algorithm to build a local dynamics graph, (3) a simple exploration procedure based on the local dynamics graph.


**Summary Of The Review:**

Overall, I am not convinced that the method is truly useful in real settings beyond the deterministic + spatial problems that the paper focuses on. The formalization of equivalent action sequences satisfied my curiosity but is perhaps slightly overkill and unnecessary.

---

> ### Author Response · Authors · 2021-11-23
> **New experiment in a stochastic setting**
>
> We thank the reviewer for their comments. We provide additional remarks and clarifications in what follows.
>
> Q: The authors also discuss an extension to the stochastic case, where equivalences are determined based on next state distributions. This seems nearly impossible in the model-free RL setting.
>
> A: In a very general stochastic setting, it might be hard to have good priors on action sequence equivalences. Nevertheless, our method can still be used when priors are available. To showcase this, we modified the setting of Freeway (Sec. 5.4) to add “sticky actions” (setting from[2]): with probability 0.25, the next action is identical to the previous action. The environment is thus stochastic, although our previous priors remain approximately true. Training with the exact same hyperparameters as in the deterministic setting, we obtain similar results. We updated Fig 4.c using this new setting.
>
> Q: All experiments are performed on simple domains. It is unclear how applicable this method is in real problems.
>
> A: Our framework is explicitly built for discrete control problems, and we believe that a lot of complex problems can be formalized in this setting (e.g. the Atari suite). Freeway for example is a quite complex problem, due to the large observation space, and cars arriving randomly (and now stochastic dynamics due to sticky actions).
>
> [2]: Revisiting the Arcade Learning Environment: Evaluation Protocols and Open Problems for General Agents, Machado et al., IJCAI 2018.

---

> > ### Comment · Reviewer_Ln9a · 2021-11-28
> > **Response**
> >
> > Thank you to the authors for providing the response. The new experiment on sticky actions partially addresses the concern on stochastic problems, but for general problems this seems to still be very challenging. In any case, I'd like to increase my score slightly.
> >
> > I believe the idea of the paper is interesting but am still not sure that the results are generalizable to real problems. To significantly improve the paper (esp. for a conference like ICLR), I believe some of formal setup could be replaced by more extensive experimental work to really prove the idea works.

---

### Author Response · Authors · 2021-11-23
**Response to all reviewers**

We would like to thank all the reviewers for the time spent on our paper, and for the constructive comments provided. We have revised our submission in the ways detailed in our answers below. To make way for additions to the paper we removed some of the detail from section 4.1. We hope the updated version addresses your concerns.

---

### Decision · Program_Chairs · 2022-01-20

**Decision:**

Reject

**Comment:**

This well-written paper introduces an improved exploration strategy by exploiting knowledge about sequences of actions that lead to the same state. The idea is straightforward and easy to understand and apply, which makes it potentially interesting. An important downside is the limited applicability of the method, as there mainly seems to be an advantage in (mostly deterministic) grid-like MDPs. In addition, priors about action-sequence equivalences have to be available. Overall, the contribution of the paper is not deemed significant enough for publication at a top-tier conference like ICLR by the majority of the reviewers as well as myself. For these reasons, I recommend rejection.